# New Building Blocks for Self-Healing Polymers

**DOI:** 10.3390/polym14245394

**Published:** 2022-12-09

**Authors:** Elena Platonova, Polina Ponomareva, Zalina Lokiaeva, Alexander Pavlov, Vladimir Nelyub, Alexander Polezhaev

**Affiliations:** 1Center NTI “Digital Materials Science: New Materials and Substances”, Bauman Moscow State Technical University, 2nd Baumanskaya Str., 5/1, 105005 Moscow, Russia; 2Physics Department, Lomonosov Moscow State University, Leninskie Gory 1-2, 119991 Moscow, Russia; 3Center for the Study of Molecular Structure, A.N. Nesmeyanov Institute of Organoelement Compounds, Vavilova Str., 28, 119334 Moscow, Russia

**Keywords:** mechanical properties, Diels–Alder reaction, polyurethane, self-healing, thermal properties

## Abstract

The healing efficiency in self-healing materials is bound by the ability to form blends between the prepolymer and curing agent. One of the problems in the development of self-healing polymers is the reduced affinity of the bismaleimide curing agent for the elastomeric furan-containing matrix. Even when stoichiometric amounts of both components are applied, incompatibility of components can significantly reduce the effectiveness of self-healing, and lead to undesirable side effects, such as crystallization of the curing agent, in the thickness and on the surface. This is exactly what we have seen in the development of linear and cross-linked PUs using BMI as a hardener. In this work, we present a new series of the di- and tetrafuranic isocyanate-related ureas—promising curing agents for the development of polyurethanes-like self-healing materials via the Diels–Alder reaction. The commonly used isocyanates (4,4′-Methylene diphenyl diisocyanate, MDI; 2,4-Tolylene diisocyanate, TDI; and Hexamethylene diisocyanate, HDI) and furfurylamine, difurfurylamine, and furfuryl alcohol (derived from biorenewables) as furanic compounds were utilized for synthesis. The remendable polyurethane for testing was synthesized from a maleimide-terminated prepolymer and one of the T-series urea. Self-healing properties were investigated by thermal analysis. Molecular mass was determined by gel permeation chromatography. The properties of the new polymer were compared with polyurethane from a furan-terminated analog. Visual tests showed that the obtained material has thermally induced self-healing abilities. Resulting polyurethane (PU) has a rather low fusing point and thus may be used as potential material for Fused Deposition Modeling (FDM) 3D printing.

## 1. Introduction

Polyurethanes (PU) are a class of widely used polymers, which are widely applied, from shoemaking to constructional and composite materials [1,2,3]. Their share is almost 8% of the world’s plastic production and they are in the 6th place among the most utilized polymers. The versatility of polyurethanes, which have a wide range of possible mechanical properties, from hard plastics to highly elastic rubber-like materials, allows them to replace both synthetic (polyvinyl chloride, rubber, polystyrene) and natural (leather) materials [4,5]. The majority of commercially produced PUs are not recyclable and are commonly not recyclable and are incinerated or landfilled. Almost 50% of post-consumer or postproduction PU wastes are utilized that way [6]. Mechanical treatment (regrinding into powder and using as filler) [7] or burning [8,9] are most commonly used as recycling methods. In some cases, PUs can be recycled by catalytic glycolysis. Recycled glycols, available for reuse in polymer preparation, are formed during that process [10]. However, chemical recycling requires more resources in terms of energy costs and additional chemicals compared to mechanical ones [11]. Fused deposition modeling (FDM) 3D printing is a powerful tool that enables the fabrication of flexible details of intricate shapes from different materials—metals, thermoplastics, and composites [12]. Thermoplastic polyurethanes are promising materials in 3D printing applications, but unfortunately thermoreactive ones are not suitable for the method. Cross-linked polymers with dynamic cross-links (for example, the ones based on the thermoreversible Diels–Alder reaction (DA reaction)) are suitable for FDM 3D printing [12,13].

As all materials, polyurethanes suffer from damage and microcracks that may dramatically affect the product lifespan and eventually result in polymer degradation and failure [14]. Scientific research in that area was focused on the development of materials with enhanced reliability or non-destructive analytical methods. Self-healing polymers, which can cure occasional damage as living tissue, opened a new way to create more long-lasting and environmentally friendly materials [15].

At first, so-called extrinsic self-healing polymer systems were developed. Curing in this type of material was achieved by the introduction of microcapsules or microcapillaries with liquid monomer and particles of a catalyst into the polymer matrix. Upon the occurrence of a microcrack, monomer from the capsule releases, and the polymerization reaction proceeds and the damage is healed [16,17]. The most obvious disadvantage of such polymer systems is a limited number of curing efforts. Such composites lose their self-healing properties when resources of the monomer and catalyst are exhausted [18].

The other large class of remendable polymers is intrinsic systems. They do not require the introduction of additional material, and self-healing in such systems occurs due to the specific properties of the polymer matrix, such as thermoplasticity or reversible interactions (formation of disulfide bridges, hydrogen bonds, metal complexes, or covalent bonds via the Diels–Alder reaction), but at the same time requires triggers (heat, for example) to initiate the healing process. [19]. The intrinsic self-healing approach is widely used for the design of novel self-healing PUs. The most common reversible Diels–Alder reaction between furanic and maleimide groups applied for the design of thermally remeandable PU with self-healing properties due to predictable formation and breaking of strong covalent C-C bonds resulted in robust materials with good mechanical properties [20]. Furans and maleimides may be incorporated in the polymer chain (in the backbone or as the side groups) or commercially available maleimides could act as a reversible curing agent for furan-containing oligomers. Self-healing usually requires thermocycling to first partially cleave Diels–Alder adducts (DA adducts) and let a crack shrink and disappear due to plasticity at elevated temperature; after cooling, the DA reaction takes place and restores the hardiness of the material.

However, the majority of known PUs based on the DA reaction suffer from the incomplete restoration of mechanical properties that could be the result of either the rather low efficiency of the mass-transfer process or furanic groups scavenging in hard domains of the PU, so the DA reaction with the maleimide may be challenging [21]. It is well known that PUs are biphasic materials, where urethane-containing fragments form hard domains and aliphatic ethers or esters soft and elastic ones. To facilitate the DA reaction, it is important to concentrate maleimide and furanic groups in the same domains. We faced a similar problem in one of our previous works on the synthesis of the linear remendable PU [22].

The principle “like dissolves like” may appear as a possible resolution for that problem. To enhance the prepolymer-to-curing agent affinity, it is worthwhile to use isocyanate-derived curing agents, which can be easily mixed with PU prepolymers and promote accessibility of the furanic and maleimide groups for the DA reaction. This method may enlarge the self-healing efficiency and additionally reinforce the resulting material (via hydrogen bond formation between prepolymer hard segments and urea fragments of the curing agent) as well. A rather large number of papers on the topic of self-healing polyurethanes based on the DA reaction commonly used commercially available bismaleimide ((1,10-(methylenedi-1,4-phenylene)bismaleimide, BMI) as a reversible curing agent for a variety of furan-derived elastomers or oligomers [23,24,25]. To the best of our knowledge, we find few reports that explore a backward possibility to apply relatively small di- or oligofuranic molecules to maleimide-derived thermoplastics [26,27,28]. To conclude, we present here a series of small furan-based curing agents for self-healing PUs derived from the most common isocyanates (MDI, TD, and HDI). We explore the self-healing performance of compositions based on a reversed approach and compare them with classical ones. We intend to increase the efficiency of the DA process by incorporating furans and maleimides into the same domain of the polymer. HEMI (N-(2-hydroxyethyl)-maleimide)-terminated PU prepolymer (PU-HEMI) and difuranic curing agent, derived from TDI and furfurylamine, were used for preliminary hypothesis testing (Figure 1).

The structure and properties as well as the self-healing ability of the resulting material were investigated. A complete study describing the synthesis and properties of the materials based on all furan-based curing agents synthesized will follow.

## 2. Materials and Methods

### 2.1. Materials

DFA (difurfurylamine) was prepared by a modified literature method [29]. N-(2-Hydroxyethyl)-maleimide (HEMI) was synthesized via reported procedure [30]. MDI (methylene diphenyl diisocyanate, 98%), HDI (hexamethylene diisocyanate, 99%), BMI (1,10-(methylenedi-1,4-phenylene)bismaleimide, 95%), and Tin(II) 2-ethylhexanoate (Sn(Oct)_2_) were purchased from Sigma-Aldrich (Steinheim, Germany) and used as received. TDI (toluene-2,4-diisocyanate, 80% of 2,4-isomer) was purchased from Sigma-Aldrich (Steinheim, Germany) distilled in a vacuum, and frozen out to separate the 2,6-isomer. FOH (furfuryl alcohol, 98%) was purchased from Sigma-Aldrich (Steinheim, Germany) and distilled under reduced pressure to remove tarry material. FA (furfurylamine, 99%) was purchased from Acros (Geel, Belgium) dried over NaOH, and distilled in a vacuum prior to use. PPG2000 (polypropylene glycol, Mn = 2000) was purchased from Sigma-Aldrich (Steinheim, Germany) and dried under vacuum at 110 °C prior to use. DMF (N,N-dimethylformamide) was purchased from Acros (Geel, Belgium) dried over CaH_2_, and distilled in vacuum prior to use. Methylene chloride (CH_2_Cl_2_) was purchased from Acros (Geel, Belgium) dried over P_2_O_5_, and distilled.

Detailed instructions on the preparation of furanic curing agents and remendable polyurethanes and further characterizations can be found in the “Appendix A”.

### 2.2. Characterization and Measurement

NMR spectra were recorded by a Bruker Avance 600 NMR Spectrometer (Billerica, MA, USA) (600.1 MHz); using the residual proton signal of a deuterated solvent as a reference, chemical shifts were reported as parts per million downfield from tetramethylsilane (TMS). ATI-FTIR was performed on a Nicolet iS10 spectrometer (Waltham, MA, USA) in the range of 4000 to 650 cm^−1^ on a germanium crystal. The thermal behavior was examined by DSC, with a NETZSCH DSC 204 F1 Phoenix (Selb, Germany) within a temperature range of −80 to 180 °C at a heating/cooling rates of 10 K min^−1^ in argon atmosphere. A sample weight of about 18 mg was used for measurement. Thermogravimetric analysis (TGA) was performed on NETZSCH TG 209 F1 Libra (Selb, Germany) within a temperature range of 30 to 550 °C at a heating/cooling rate of 10 K min^−1^ in an argon atmosphere. Gel permeation chromatography (GPC) analyses were performed using a Shimadzu Prominence LC-20 (Kyoto, Japan) apparatus equipped with a RID 20A refractive index detector and three PSS GRAM analytical columns 100 Å (8.0 × 300 mm^2^) linked in series, thermostatted at 40 °C, and using N,N-dimethylformamide (DMF) containing 0.5 g/L LiBr as the mobile phase at a flow rate of 1.0 mL min^−1^. Molecular weights were estimated against polymethylmethacrylate (PMMA) standards. Mechanical properties (σ_t_, E, ε, σ_t_ (ε)) were determined from the uniaxial tensile strength test. Dogbone-shaped samples were cut from film using a special blade (length 15 mm, width 2 mm, see Appendix A) and hand press. Film thickness was from 1.5 to 2.0 mm. The samples were stretched on a Zwick Roell Z2.5 universal testing machine (Ulm, Germany) at a 150 mm/min loading speed. An applied load P and a strain ε were recorded as loading diagrams. Further, the stress at break σ_t_, the elongation ε, and the elastic modulus E were calculated from the diagrams.

## 3. Results

### 3.1. Preparation Process for Furanic Curing Agents

Isocyanates react cleanly and completely with alcohols, primary and secondary amines. Furyl alcohol and furfurylamine are the large-scale available and inexpensive source of furan moieties, with an advantage of being bioderived [31]. Our synthetic approach shown in Figure 1, based on a robust reaction, resulted in almost quantitative yields of desired products of a high purity without additional purification. Difurylamine, furfurylamine, and furanic alcohol reacted with most common industrial isocyanates (HDI, TDI and MDI) with formation of compounds of the H, T, and M series, respectively. The structures of the furanic curing agents were established by NMR and IR spectroscopy. The characteristic band of the isocyanate (2270 cm^−1^) in the reaction mixture disappears and a new band of the carbonyl group at ~1630 cm^−1^ emerges. In the NMR spectra of the reaction products, the chemical shifts of furanic protons shifted in a weak field (from 7.36 ppm in furylamines and furyl alcohol to ~7.60 ppm in furan-containing ureas). Resulting curing agents are colorless or light-yellow powders, soluble in polar solvents (DMF and DMSO) and insoluble in dichloromethane, hexane, and acetone. Melting points for all furan-ureas were measured in a vacuum in a sealed capillary and via the DSC method (see Appendix A). It is interesting to note that the melting points of difurylamine and furyl alcohol derivatives in the series are rather close, while furylamine derivatives have substantially higher melting points (in the case of curing agents of the T and H series, the difference is almost 100 °C). We presume that such an unusual thermal behavior could be associated with higher content of hydrogen bonds between urethane fragments in curing agents: alcohol and diamine derivatives have two secondary amino groups per molecule, and amine derivatives have four secondary amino groups per molecule.

### 3.2. Structures, Properties, and Thermal Remendability of Self-Healing Polyurethanes

Previously, we and other research groups obtained linear polymers from BMI and furan-terminated prepolymers via a general two-step method. In the first stage, NCO-terminated prepolymer was synthesized by the interaction of polyol and isocyanate; at the second step, this prepolymer was functionalized with furylamine or furfuryl alcohol [22,32,33,34,35,36]. In this work, we used a similar approach, but with reversed functionality of reagents. Linear prepolymer PU-HEMI was synthesized by a modified method [22] (Figure 2). The general two-step method was used: at the first stage, we obtained prepolymer with terminal isocyanate groups from glycol PPG-2000 (soft segment) and TDI (hard segment); at the second stage, it reacted with HEMI. PU-HEMI was characterized by NMR and IR (see Appendix A). For the proof of the concept, we chose the FA-T curing agent of the T series. The choice of the curing agent is guided by a goal to compare the properties of the PUs synthesized by standard and modified methods.

A visual investigation of the thermal remendability of remendable polyurethane (PU-DA) is shown in Figure 2.

The resulting polymer PU-DA was also characterized by NMR and IR spectroscopy (for detailed NMR spectra, see Appendix A). Identification and comparative analysis of structure for prepolymer and remendable PU are given below.

In the ^1^H spectra of the PU-HEMI (Figure 3), signals of aromatic protons of TDI are overlaid with signals of double-bond protons of the maleimide cycle: intensity of those signals at 7.04–7.05 ppm decreased during the curing process, and this fact suggests successful DA adducts formation. Additionally, a notable decrease in the signals of unbound furanic groups at 7.59, 6.40, and 6.27 was detected, which furthermore indicates the interaction between furans of the curing agent and maleimides of the prepolymer.

A comparison of the IR spectra was more illustrative. Bands of the asymmetric vibration of C=O groups conjugated with the maleimide cycle in prepolymer (1716 cm^−1^) and PU-DA (1703 cm^−1^) spectra were preserved. A small red shift of the band may be caused by the breaking of the C=O and maleimide heterocycle conjugation via the DA reaction occurring [37]. On the other hand, bands of the double bond (829 cm^−1^) and deformational vibration of the cycle (696 cm^−1^) in maleimide completely disappeared, which indicates DA reaction occurrence [38]. The disappearance of the bands at 884 and 739 cm^−1^ associated with the furanic ring also proves the cross-linked via DA reaction PU formation [39].

The thermal properties of the PU-DA polymer were investigated with the TGA method (Figure 4). Thermal stability of the material was a little bit lower than for the linear PUs synthesized previously: T_5%_, T_10%_, and T_max_ were 251, 280, and 370 °C, respectively (for linear PU cured with BMI, those were 288, 312, and 380 °C, respectively) [25]. We assume that DA adducts decompose to initial components via the rDA reaction and that the thermal stability of furylurethanes used for curing in this work is lower than that of BMI.

Additionally, thermal properties and DA-bond reversibility were investigated by the DSC method. The first heating curve of PU-DA (Figure 4) demonstrated a broadened endothermal peak at 120 °C corresponding to the decomposition of DA adducts. The absence of such a signal on the second heat curve showed the completeness of the rDA-reaction in the polymer sample. Both thermograms of the prepolymer and PU-DA demonstrated the glass transition process only for the soft segment of polyurethane; Tg for prepolymer was −45.3 °C and for PU-DA it was −48.4 °C (see Appendix A). Those results were consistent with the glass transition of the PPG-2000-derived PUs [22,36].

There were a few reports on linear self-healing PU that reported a molecular weight of Mn = 27,000 and PDI = 1.88 (soft segment—PPG2000, measured in THF) [36], and Mn = 18,557 and PDI = 2.48 (soft segment—PBA1000, measured in DMF) [40]. However, the difference in column, solvents, and standards used makes comparisons of Mw of the material impossible. To make a valid comparison of classical approaches and our approaches to linear self-healing PU (to perform polymerization with the same polyol and isocyanate (PPG-2000 and TDI), under the same conditions and evaluate Mw and Mn by the same protocol), we synthesized the linear PU (PU-DA-FA) from furylamine-functionalized prepolymer cured by BMI using a known protocol (Figure 3). The results (Table 1) show that the proposed reversed approach resulted in a four-times-higher Mw of the PU-DA compared to PU-DA-FA under the same reaction conditions. This makes the reversible approach more promising compared to the classical one, and the more detailed investigation of it is ongoing.

We also performed a preliminary evaluation of the self-healing efficiency of PU-DA. For quantification of self-healing process efficiency, the measurement of elastic modulus and tensile strength was performed. As a control measurement, five samples from the first group were tested for tension. The samples from the second group were damaged with a blade in the middle. Damaged samples were subjected to thermal treatment: 2 h at 120 °C and 4 days at 60 °C. After completion of the healing process, the samples from the second group were tested for tension in the same way as the first (control) group. For quantification of elastic modulus in each group of samples, the mean load curve was used (average curve for all samples of the group). Representative loading diagrams for pristine and healed specimens of PU-DA are shown in Figure 5 (details of the experiment and calculations are provided in the Appendix A). Tensile properties are listed in Table 2.

Upon loading, the PU-DA sample reacted similarly to the linear remendable PU we obtained via the classic approach previously [22]. PU synthesized via the modified method was inferior in tensile properties to classic PU (values of E and σ_t_ for classic PU were 40.0 and 25.8 MPa correspondingly), but it exceeded classic PU in self-healing efficiency: the value of η_E_ was 100%, and η_σ_ was 93 % (for classic PU, the same values were 95 and 54%, correspondingly).

## 4. Conclusions

A series of the new tetra- and difuranic compounds derived from commonly used isocyanates were synthesized, and their ability to act as a curing agent in self-healing compositions with HEMI-PU was investigated. PU specimens showed thermal-induced self-healing ability based on the DA reaction, which was confirmed by DSC, IR, and NMR. Further work is being undertaken to investigate PU properties, optimal composition, and determination of the self-healing efficiency. Furanurethanes obtained could be used as curing agents for PU elastomers as well as comonomers in combination with corresponding bismaleimides for the synthesis of new polymeric materials via the Diels–Alder reaction. The reversed approach was shown to be more promising than the classical one in the formation of high Mw polymers with faster self-healing.

Later, we are planning to obtain PUs from other urea-derived curing agents, to investigate their properties and self-healing efficiency.

We suppose that obtained remendable polyurethane could melt due to thermally reversible bonds in DA adducts at approximately 80 °C, and we plan to apply it for development of recyclable FDM 3D printing material.

## Data Availability

The data presented in this study are available on request from the corresponding author.

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
