# Peer review of "New Building Blocks for Self-Healing Polymers"

_polymers, 2022, doi:10.3390/polym14245394_

Round 1
Reviewer 1 Report
Journal: Polymers (ISSN 2073-4360)
Manuscript ID: polymers-2071942
Title: New building-blocks for self-healing polymers
The authors presented a new series of the di- and tetrafuranic isocyanate-related ureas – promising curing agents for the development of polyurethanes-like self-healing materials via Diels-Alder reaction. The commonly used isocyanates (4,4′-Methylene diphenyl diisocyanate, MDI; 2,4-Tolylene diisocyanate, TDI; and Hexamethylene diisocyanate, HDI) and furfurylamine, difurfurylamine, and furfuryl alcohol (derived from biorenewables) as furanic compounds were utilized for synthesis. The remendable polyurethane for testing was synthesized from a maleimide-terminated prepolymer and one of the T-series urea. Self-healing properties were investigated by thermal analysis. Molecular mass was determined by gel permeation chromatography. The properties of the new polymer were compared with polyurethane from furan-terminated analog. Visual tests showed that the obtained material has thermally-induced self-healing abilities. Resulting PU has a rather low fusing point and thus may be used as potential material for FDM 3D printing.
The paper will be ready for publication after major revision.
The methodology section describes each of the methods in depth such that the experiments could be reproduced by another researcher.
The experimental design, analytical methods and interpretation of the results are very good.
The authors need to interpret the meanings of the variables.
The abstract should be rewritten to reflect the significance of the proposed work.
Please highlight your contributions in introduction.
“Figure 3. NMR(a) and IR (b) spectra of prepolymer, PU-DA and FA-T.”, please enlarge fonts. Put subfigure in vertical configuration instead of horizontal configuration.
What is the manufacture of “Bruker Avance 600 NMR Spectromete”?
What is the manufacture of “DSC, with a NETZSCH DSC 204 F1 128 Phoenix”?
“Linear prepolymer PU-HEMI was synthesized by a modified method”, Explain this method in details.
The introduction should be supported by recent publications to show the importance of composite materials in different engineering applications such as in energy harvesting especially from MDPI:
Bistable Morphing Composites for Energy-Harvesting Applications
Future work must be included.
Looking and wishes for the revised version.
Author Response
The authors need to interpret the meanings of the variables.
We agreed with this comment, changes were made.
The abstract should be rewritten to reflect the significance of the proposed work.
We agreed with this comment, changes were made.
Please highlight your contributions in introduction.
Contributions are listed in Author Contributions.
“Figure 3. NMR(a) and IR (b) spectra of prepolymer, PU-DA and FA-T.”, please enlarge fonts.
We agreed with this comment, changes were made.
Put subfigure in vertical configuration instead of horizontal configuration.
The horizontal configuration of Figures was used by Instructions for Authors of The Polymers MDPI Journal.
What is the manufacture of “Bruker Avance 600 NMR Spectromete”?
We agreed with this comment, changes were made.
What is the manufacture of “DSC, with a NETZSCH DSC 204 F1 128 Phoenix”?
We agreed with this comment, changes were made.
“Linear prepolymer PU-HEMI was synthesized by a modified method”, Explain this method in details.
The detailed synthetic procedure explained in Supporting information in 2.10 Synthesis of linear polyurethane PU-HEMI.
The introduction should be supported by recent publications to show the importance of composite materials in different engineering applications such as in energy harvesting especially from MDPI:
Bistable Morphing Composites for Energy-Harvesting Applications
We agreed with this comment, changes were made.
Future work must be included.
We agreed with this comment, changes were made.
Reviewer 2 Report
In this work, the authors introduced multiple promising curing agents for the development of polyurethanes-like self healing materials via the Diels-Alder reaction. Thermally-induced self-healing properties, and molecular weight were then investigated and confirmed. Based on these features, the authors claimed can be used as a potential substrate for FDM 3D printing. There are some concerns about experiment designs. I would like to suggest a major revision before acceptance.
Comment:
1. For self-healing properties, only surface cutting, and healing are insufficient. Other properties like mechanical performance before and after cracks and adhesive performance are necessary. Please supplement these characterizations.
2. In this work, authors chose fused deposition modeling (FDM) 3D printing as a potential use. Please give some printed examples based on the polyurethanes-like self-healing materials developed herein.
3. Why did the authors choose fused deposition modeling (FDM) 3D printing as the specific application? How about other kinds of 3D printing technologies, such as direct ink printing?
Author Response
- For self-healing properties, only surface cutting, and healing are insufficient. Other properties like mechanical performance before and after cracks and adhesive performance are necessary. Please supplement these characterizations.
We agreed with this comment, changes were made.
- In this work, authors chose fused deposition modeling (FDM) 3D printing as a potential use. Please give some printed examples based on the polyurethanes-like self-healing materials developed herein.
At present moment we investigate materials derived from other urea-containing curing agents and make adjustments to the polymer composition properties (fusing temperature and self-healing efficiency). After estimating the optimal composition of the PU we plan to develop the filament for 3D printing.
- Why did the authors choose fused deposition modeling (FDM) 3D printing as the specific application? How about other kinds of 3D printing technologies, such as direct ink printing?
In direct ink printing technology pastes made from polymers and highly volatile solvents are used. The solvent is a fundamental component of DIW inks, necessary to give fluidity to the system during the extrusion and deposition processes. Organic solvents can be flammable, toxic, or leave residues that hinder the electrochemical performance of the printed devices. Our polymeric materials are soluble only in DMF, DMSO (difficultly volatile solvents), and THF. The last one is a highly volatile solvent, but it is highly flammable and may cause the corruption of the printing device.
Round 2
Reviewer 1 Report
Accept.
Reviewer 2 Report
The current version is suitable for publication.